# Pharmacokinetics and Clinical Implications of Two Non-Tablet Oral Formulations of L-Thyroxine in Patients with Hypothyroidism

**DOI:** 10.3390/jcm11123479

**Published:** 2022-06-16

**Authors:** Pierpaolo Trimboli, Stéphane Mouly

**Affiliations:** 1Servizio di Endocrinologia e Diabetologia, Ente Ospedaliero Cantonale (EOC), 6500 Bellinzona, Switzerland; pierpaolo.trimboli@eoc.ch; 2Facoltà di Scienze Biomediche, Università della Svizzera Italiana (USI), 6900 Lugano, Switzerland; 3Assistance Publique-Hôpitaux de Paris Nord, Département Médico-Universitaire (DMU) INVICTUS, Département de Médecine Interne, Hôpital Lariboisière, INSERM UMRS-1144, Université de Paris Cité, 75010 Paris, France

**Keywords:** levothyroxine, hypothyroidism, soft-gel capsule, oral solution, pharmacokinetics, drug–food interactions, gastrointestinal conditions, switch

## Abstract

Background: Increased knowledge of the pharmacokinetic characteristics of orally administered levothyroxine (L-T_4_) has improved individualization of dosing regimens. However, up to 40–45% of patients, depending on the leading cause of hypothyroidism, are still over- or, more often, undertreated. Unintentional non-adherence to L-T_4_ replacement therapy includes all situations of unintended drug–drug and drug–food interactions as well as fasting conditions that are not necessarily respected by patients. Results: In this specific context, the overall information concerning those factors with the potential to affect L-T_4_ absorption refers only to tablet formulation. Indeed, this is the reason why new non-tablet formulations of L-T_4_ were introduced some years ago. In this regard, the current literature review was designed to summarize pharmacokinetic, drug and food interactions and clinical data focusing on two new oral L-T_4_ formulations, i.e., liquid and soft-gel capsule in healthy volunteers and patients with primary hypothyroidism. The non-tablet L-T_4_ soft-gel capsules and solution have proven bioequivalence with the usual L-T_4_ tablet Princeps and generic formulations. Clinical studies have suggested higher performance of non-tablet formulations than tablet in those patients with suboptimal adherence. The impact of gastrointestinal conditions and variation of gastric pH was lower with either soft gel/solution than with tablets. In addition, the extent of drug–drug and drug–food interactions remains low and of uncertain clinical relevance. Conclusions: Pending further studies allowing one to extend the use of soft-gel/solution preparations in unselected patients, non-tablet L-T_4_ formulations should be considered as a first-line choice, especially in those patients with moderate-to-high potential of suboptimal tablet performance.

## 1. Introduction

Thyroid hormones play a critical role in the human body and, as such, if such hormones are absent, they must be replaced. Levothyroxine (L-T_4_) is the recommended replacement thyroid hormone for both primary and central hypothyroidism. Since the 1970s, most hypothyroid subjects worldwide have been treated with the usual oral tablet L-T_4_; thus, it is in the first decile of most prescribed medications in Western countries [1,2]. Levothyroxine is a critical-dose drug since even slight variations in blood concentration may determine treatment failure (i.e., incomplete replacement or iatrogenic thyrotoxicosis). Then, individualized L-T_4_ treatment is required and a close clinical follow-up to avoid under- and overtreatment of patients.

The daily dose of L-T_4_ depends on the leading cause of hypothyroidism (mostly primary hypothyroidism) as well as on several potential determinants, mainly patients’ lean body mass, which is not routinely measured in clinical practice, and the therapeutic goal (i.e., replacement or Thyreo-Stimulating Hormone (TSH)-suppressive therapy in thyroid cancer patients, when needed). Recent improvement in knowledge concerning the pharmacokinetic characteristics of orally administered L-T_4_ has improved individualization of dosing regimens. However, up to 40–45% of patients, depending on the leading cause of hypothyroidism, are still over- or, more often, undertreated [3], and nonadherence only accounts for less than 17% of the cause of poorly targeted dosing regimens [1,2]. Unintentional non-adherence to L-T_4_ replacement therapy includes all situations of unintended drug–drug and drug–food interactions, as well as fasting conditions that are not necessarily respected by patients despite it being well known that undergoing L-T_4_ replacement therapy in fasting conditions improves intestinal absorption of the active ingredient [4]. Besides these causes of increased doses of L-T_4_, some conditions and disorders may enhance daily L-T_4_ requirement by affecting its intestinal absorption, including celiac disease, lactose intolerance and infections, especially *Giardia intestinalis* infection [2,4]. In this specific context, it has to be remarked that the overall information concerning those factors with the potential to affect L-T_4_ absorption refers only to tablet formulation. Indeed, this is the reason why new non-tablet formulations of L-T_4_ were introduced some years ago. Furthermore, numerous studies have been published during the last decade in this field and reported that hypothyroid patients with not-on-target TSH for tablet L-T_4_ can experience optimal (or at least improved) thyroid hormonal balance after switching to liquid or soft-gel L-T_4_ formulation despite an unchanged dose. In contrast, to the best of our knowledge, no papers have recorded an improvement of control of hypothyroidism when a patient switches from soft gel or liquid to tablet L-T_4_.

Alongside this increasing information, whether such improvement in dosing regimens and pharmacokinetic stability is or is not associated with substantial improvement in clinical symptoms and number and frequency of both over- and undertreated patients remains debated. In this regard, the current literature review was designed to summarize pharmacokinetics, drug and food interactions and clinical data focusing on two new L-T_4_ preparations, i.e., liquid and soft-gel capsule, in healthy volunteers and patients with primary hypothyroidism.

## 2. General and Pharmacological Considerations

Around 80 ug/day of L-T_4_ is produced in physiological conditions and L-T_4_ circulating concentration is up to 60-fold higher than L-T_3_ [1,2]. Numerous L-T_4_ oral formulations are currently available for the treatment of hypothyroidism [5]. Excipients vary among these different formulations, which may explain the difficulty of switching from one to another formulation, as observed in several previously reported clinical cases [6,7,8,9]. Liquid oral formulations usually contain glycerol and ethanol or glycerol alone, while soft-gel capsules contain gelatin, glycerol and/or purified water [10]. Sodium L-T_4_ is considered identical to the physiological hormone. L-T_4_ is mainly absorbed through the distal small intestine, mainly jejunum and ileum, by means of organic anion-transporting polypeptide (OATP), mainly the OATP1A2, OATP1B1 and OATP1C1 isoforms. Oral bioavailability is 70–80% in euthyroid persons but depend on release from the formulation, dissolution and transporter-dependent intestinal absorption. The maximum concentration is reached in 1.5–3 h [2,4,5,11,12]. L-T_4_ is then distributed in the body with a limited volume of distribution, ranging from 11.6 L in euthyroid volunteers and up to 15 L in primary hypothyroid subjects, presumably due to as much as 99.8% protein binding in the blood. Tissue distribution is also dependent on cellular uptake, activation by deiodination of L-T_4_ into L-T_3_, receptor binding and the degree of liver glucuronidation and kidney sulfation. L-T_4_ is mainly cleared through the kidney (80% of the total clearance) and renal clearance is decreased by 40% in hypothyroid patients versus healthy control volunteers (0.038 versus 0.055 L/h, respectively) [2,4,5,11,12]. However, L-T_4_ half-life is roughly similar between euthyroid subjects and hypothyroid patients, ranging between 6 and 7.5 days, while the L-T_3_ half-life ranges between 1 and 1.4 days in euthyroid subjects and hypothyroid patients.

L-T_4_ dosing regimen in hypothyroid patients is usually 1.3 to 1.6 ug/kg/day, requiring careful titration especially in the elderly. Likewise, several disease conditions and specific populations may alter pharmacokinetic parameters of L-T_4_, summarized in Table 1 [5]. In this regard, switching to a more stable formulation may require these usual conditions to be considered in clinical practice.

## 3. Bioequivalence Studies with Soft-Gel Capsule and Oral Liquid Formulation

Because levothyroxine pharmacokinetics is highly variable due to the influence of several physiological (e.g., old age, pregnancy) and pathological (liver or kidney impairment, morbid obesity, gastrointestinal disorders) conditions, as well as numerous drug–drug and drug–food interactions, reaching bioequivalence between Princeps and the numerous formulations currently marketed may appear as a challenge. Indeed, results of the pharmacokinetic studies involving L-T_4_ need to be carefully interpreted, considering baseline hormone concentration and central nervous system feedback mechanisms. In this regard, pharmacokinetic studies have focused on L-T_4_ serum concentrations rather than TSH, the target biomarker in clinical practice.

Numerous studies, most of them conducted in healthy euthyroid subjects, have confirmed bioequivalence between, separately, the classical sodium levothyroxine formulation and L-T_4_ soft-gel capsule or the latter and the solution. Overall rate and extent of exposure, as well as elimination half-life (ranging from 6 and up to 9 days) were not statistically different among the preparations but faster (approximately 30 min earlier) onset of absorption was noted with the new oral liquid L-T_4_ versus gel caps and the Princeps tablet formulation in a few studies [13]. Low- (50 μg), medium- (100 μg) and high (200 μg)-strength soft-gel capsules of L-T_4_ were bioequivalent to the European reference tablet, showing proportionality, at a dosage of 600 μg in healthy volunteers, the latter dosage being required by many agencies in order to achieve L-T_4_ concentrations sufficiently above the endogenous base [14]. In this study, it was interesting to note that the 90% confidence interval (CI) to demonstrate bioequivalence was set at 80–125%, and that the authors observed as much as 15% intra-individual variability in their study enrolling only healthy subjects despite stable L-T_3_ concentrations and similar tolerance between the respective formulations [14]. Two distinct soft-gel capsules (the “old” one and the “new” one manufactured using the new Food & Drug Administration Potency guidelines) were shown to be bioequivalent with each other and with the American tablet formulation (Synthroid), using bioequivalence 90% with a confidence interval of 80–125% for mean ratio of the serum L-T_4_ systemic exposure [15]. Finally, bioequivalence, as assessed by using the 90% CI for mean ratio of the systemic exposure (i.e., C_max_ and AUC_0-48_) was also confirmed between the two new oral formulations, oral solution and soft-gel capsules, in a recent crossover study conducted in healthy subjects taking a single oral dose of 4 × 150 μg with or without water and using current European Medicine Agency (EMA) prespecified bounds of 90.00 to 111%, as recently required for bioequivalence studies [12].

## 4. Switching from the Usual Tablet to Solution or Soft-Gel Capsule Formulations in Clinical Practice

Levothyroxine changes, either dose titration or preparation, occur frequently in patients with suboptimal TSH and/or persistent symptomatology. The therapeutic goal in hypothyroidism is to replace the thyroid function with neither over- nor under substitution, starting with an estimated ideal dose of L-T_4_ able to obtain normal TSH and normal or slightly higher serum thyroxine. Some patients face difficulties in achieving the optimal hormonal profile concerning replacement therapy with a consequent impairment in psychological well-being in addition to other clinical symptoms suggestive of under- or over-substitution [16]. Then, changing the L-T_4_ dose and/or formulations may be common when use of healthcare sources increases, leading to poor clinical outcomes. In the CONTROL Surveillance Project conducted in 925 patients on L-T_4_ (94% for >2 years), 23.4% of respondents experienced one L-T_4_ dose change in the prior 12 months and another 8% had two or more dose changes in the prior 12 months [17]. Indeed, there is a subset of patients whose hypothyroidism is difficult to control (biochemically and/or symptomatically) with tablet L-T_4_.

Recent studies have shown higher pharmacokinetic performance of liquid and soft-gel L-T_4_ formulations in comparison with tablet one due to better L-T_4_ absorption [5,18]. Indeed, liquid and soft-gel preparations bypass the pH-dependent dissolution phase and are less affected by binding with sequestrants in bowel lumen [18]. A recent prospective randomized open-labeled unicentric study conducted in 166 recently thyroidectomized patients with blinded analysis showed that, despite similar TSH serum levels, the oral solution improved mood state, self-perceived mental well-being and other clinical symptoms, as compared to usual tablet formulation [19]. In a smaller clinical study involving only 18 patients without intestinal malabsorption, switching from tablet to soft-gel capsule at an unchanged dose improved TSH and balance after 3-month treatment [20]. In this setting of patients without malabsorption, two studies by Di Donna deserve to be cited [21,22]. There, the authors enrolled a large series of patients who had undergone thyroidectomy and showed that, while L-T_4_ requirement was not significantly different between soft-gel capsules and tablet groups, TSH was significantly lower in soft gel with no significant increase in FT4 levels. Since the setting of athyreotic patients should not cause significant bias, this finding must be transferred to clinical practice and play a relevant role in those patients in whom a specific TSH target has to be achieved. In six patients with central hypothyroidism treated with the tablet formulation and with L-T_4_ serum levels outside the target range associated with clinical symptoms, a switch to oral formulation or soft-gel capsule resulted in a significant increase in L-T_4_ serum concentration, which reached the target range in more than 90% of patients, along with a significant decrease in TSH serum levels [23]. In a retrospective review of 99 randomly selected patients (CONTROL Switch Study) investigating whether switching from tablets to gel capsule L-T_4_ might reduce dose adjustments over time and improve its tolerability and efficacy, it no change was found in TSH status in 51.5% of patients after the switch, with 78% of TSH levels being within the target range [24]. In addition, 85.8% experienced ≤1 dose adjustment after the switch to gel capsules (i.e., 33.3% one dose change, 52.5% no change), with a mean dose adjustment per patient of 1.61 ± 0.96 with tablets and 0.73 ± 0.96 with gel capsules (i.e., 54.7% decrease); 61.6% had improvement in symptoms following the switch, even if 8.1% reported worsening of symptoms [24]. In this study, the main reason for switching to a levothyroxine soft-gel capsule was the adverse effects associated with prior therapy (49 patients (59.8%)) [24]. These results were observed independently of the previous treatment before the switch (i.e., Princeps or generic tablet). These findings achieve high interest when taking into account that, as recorded in the CONTROL Surveillance Project, more than 80% of hypothyroid patients experience at least one change in their prescribed L-T_4_, with 16% reporting 5 to 10 changes and a 6.1% more than 10 changes, with no significant difference between patients with or without comorbid gastrointestinal conditions [17].

As mentioned above, bioequivalence of L-T_4_ products remains an ongoing concern, possibly because, using the usual approach, physicochemical properties of the available formulations may have been somewhat ignored. The pH-dissolution profiles of usual levothyroxine tablet formulations (i.e., Princeps and generic) were compared with that of the soft-gel capsule using an in vitro approach combined with quadruple mass spectrometry, which showed that the soft-gel capsule displayed the most consistent dissolution compared with the usual tablet formulations [25]. The levothyroxine soft-gel capsule was minimally affected by pH even in conditions where the pH is increased. Based on this in vitro study, one may assume that new oral formulations of sodium levothyroxine, especially the soft-gel capsule, may lead to more stable TSH levels in patients with disorders of the stomach and small intestine. Such disorders may include autoimmune gastritis, celiac disease, *Giardia* or *Helicobacter pylori* infection, lactose malabsorption, ulcerative colitis and even liver cirrhosis and have been associated with as much as 30% decrease in levothyroxine efficacy when using the usual tablet formulation due to reduced absorption of L-T_4_, thus requiring subsequent close monitoring of the TSH serum level and frequent dose adjustment [26,27]. In the case of symptomatic *Giardia intestinalis* infection of the intestinal tract in a 63-year-old woman previously treated with levothyroxine tablets and TSH on target before infection, a switch to an oral solution of levothyroxine because of a significant increase in TSH, at the same daily dose despite gastrointestinal symptom persistence, led to thyroid hormone normalization [27]. In this case report, the authors concluded that the reduced absorption of levothyroxine observed with the usual tablet was resolved by L-T_4_ oral solution. The key role of gastric acid in subsequent intestinal L-T_4_ absorption is proven by the increased L-T_4_ requirement found in patients with gastric disorders, as recently observed in individually tailored doses of L-T_4_ adjusted by age, body weight or body mass index [28]. In fact, better pH-related dissolution was observed in vitro for soft-gel L-T_4_ preparation than tablets, despite the demonstrated bioequivalence between the respective formulations [25]. In a recent study conducted involving 31 patients with gastric disorders and >2-year treatment with unchanged doses of L-T_4_ tablets, switching the patients to a lower dose of soft-gel capsules confirmed that there is a statistically significant lower dose requirement (−17%) to maintain stable TSH serum concentrations within the target range in more than two thirds of patients [28]. The remaining patients had associated small intestinal disorders, which may explain further dose adjustment after the switch to soft-gel capsules of L-T_4_. The impact of gastroparesis associated with diabetes I mellitus on L-T_4_ malabsorption has been previously described in another case report of a 42-year-old woman who rapidly obtained the thyroid hormonal replacement after changing to gel capsule formulation [29].

## 5. The Oral Solution and Soft-Gel Capsule Formulations Display Better Pharmacokinetic Profiles than the Usual Tablets of Levothyroxine in the Case of Drug–Drug or Drug–Food Interactions

It is known that several factors can interfere with intestinal absorption of L-T_4_, i.e., food ingestion, dietary fiber, coffee, breakfast beverages and drugs. Numerous drug–drug and drug–food interactions with L-T_4_ tablets have been identified and reports on them have been published over the past 30 years, as summarized in Table 2 [5,26]. Most of these drug–drug and drug–food interactions led to L-T_4_ intestinal malabsorption and further increase in TSH level, consistent with under-substitution despite stable dosage of L-T_4_. One of the most clinically relevant drug–drug interactions in this setting involves proton-pump inhibitors, which induce clinically significant L-T_4_ malabsorption by modifying gastric pH [30,31]. In a retrospective real-world evidence study in primary care, drug–drug interactions involving proton-pump inhibitors accounted for 88% of all drug–drug interactions reported with levothyroxine tablets, oral solution or soft-gel capsules [32]. Such interactions with proton-pump inhibitors may be solved by switching from the Princeps levothyroxine tablet to the new soft-gel capsule, as observed in a recent case report of a woman in whom an absorption test displayed a 48% increase in systemic exposure of L-T_4_ along with faster intestinal absorption [30]. In addition, despite that TSH variation in serum did not differ significantly between the tablet formulation and the, soft-gel capsule and oral solution, separately, the latter allowed more a stable dosing regimen and less dose adjustment, consistent with a more stable pharmacokinetic profile in the case of drug–drug interactions in the clinical setting [32]. In another study conducted in healthy volunteers, bioequivalence between L-T_4_ tablets and soft-gel capsules was lost when taken with esomeprazole, with a 16% decrease in the peak serum concentration and systemic exposure of L-T_4_ tablets as compared to the soft-gel capsules [31]. Calcium salt supplementation, especially in postmenopausal women, may increase the risk of L-T_4_ malabsorption and suboptimal TSH balance in hypothyroid patients. Despite that this drug–drug interaction may prevent delayed calcium salt ingestion, it is not always satisfactory as compared to switching from the tablet to the new oral solution or L-T_4_ soft-gel capsules, as recently observed in a clinical study involving 50 hypothyroid postmenopausal women taking L-T_4_ tablet therapy [33].

Current guidelines recommend taking L-T_4_ in fasting conditions, challenging adherence to medical recommendations, especially regarding other medications; a significant number of patients have suboptimal compliance with L-T_4_ ingestion as they have to postpone breakfast by at least 30 min [34]. The TICO randomized, double-blind, placebo-controlled crossover trial suggested that liquid L-T_4_ solution with ethanol should be ingested directly with or 30 min before breakfast with no statistical difference in TSH, free T_3_ and free T_4_ levels, thus potentially improving therapeutic compliance. The use of L-T_4_ liquid formulation without ethanol concomitantly with breakfast, however, remains off-label. This observation is of remarkable clinical relevance since suboptimal adherence to L-T_4_ therapy is more likely to cause TSH variation over time [34,35]. Patients undergoing treatment with L-T_4_ are usually asked to ingest the drug in the morning, at least 30 min before having breakfast, because a significantly decreased L-T_4_ absorption was reported with food when patients are treated with the usual tablet [36,37]. However, oral solution without ethanol should still be preferred in patients in whom even small changes in the free fraction of T_3_ or T_4_ may be associated with unwanted side effects, including patients with heart condition or thyroidectomy following thyroid cancer [38].

## 6. Conclusions

In summary, the non-tablet L-T_4_ soft-gel capsules and oral solution have proven bioequivalence with the usual L-T_4_ tablet Princeps and generic formulations using 90% CI% of the systemic exposure geometric mean ratios. Several clinical studies involving healthy subjects and patients have suggested higher performance of non-tablet formulations than tablet in those patients with suboptimal adherence to the correct drug ingestion. The impact of gastrointestinal conditions and variation of gastric pH has been demonstrated to be lower with either the soft-gel capsule or oral solution than with the tablet. In addition, the extent of drug–drug and drug–food interactions, including those observed with proton-pump inhibitors remains low and of uncertain clinical relevance.

Overall, despite some limitations due to the small sample size, the design heterogeneity and the inability to anticipate individual sensitivity to even small variations of TSH levels, the current review highlights that soft gel and liquid L-T_4_ formulations were proven to improve individually tailored treatment of hypothyroidism, especially in selected but substantial groups of patients (i.e., those with hypo-achlorhydria, those with polypharmacy, those who had undergone bariatric surgery, those who fed through enteric tube, patients with gastro-esophageal reflux disease and patients with *H. pylori* infection needing to be eradicated). Such problems may be overcome by using non-tablet L-T_4_ formulations. Although the field of hypothyroidis deserves further systematic studies to extend the use of these formulations in unselected patients, according to current scientific achievements, we should consider using non-tablet L-T_4_ formulations as a first-line choice, especially in those patients with moderate-to-high potential of suboptimal performance with the usual tablet as well as in those with concomitant medication increasing the risk of drug interactions and side effects, or gastro-intestinal conditions leading to decreased L-T_4_ intestinal absorption.

## Figures and Tables

**Table 1 jcm-11-03479-t001:** Pharmacokinetics of levothyroxine in special populations [5].

	Bio-Availability	De-Iodination	Protein Binding	Clearance	T_4_	T_3_
Kidney impairment		⇓	⇓			⇓
Cirrhosis		⇓	⇓		⇑	⇓
Elderly	⇓	⇓		⇓		⇓
Children				⇑	⇓	
Pregnancy				⇓	⇓	
Gastrointestinal disorders	⇓					
Food	⇓					

⇑ increase, ⇓ decrease.

**Table 2 jcm-11-03479-t002:** Impact of drug and food on levothyroxine absorption [5,26].

Interactions with Drugs	Interactions with Food
Acid suppression therapies (proton-pump inhibitors, H2 receptor antagonists, sucralfate) β-blockers Bile acid sequestrants (cholestyramine, colestipol and colesevelam) Calcium salts (carbonate, citrate, and acetate) Cation exchange resins Charcoal Chromium Ciprofloxacin Ferrous sulfate Lanthanum Multivitamins (containing ferrous sulfate or calcium carbonate) Oral bisphosphonates Orlistat Phosphate binders (sevelamer, aluminum hydroxide) Polystyrene sulfonate Rifampicin Raloxifene Simethicone Tricyclic antidepressant	Coffee Fiber Grapefruit Ingestion with a meal Papaya Soybeans and soy

## Data Availability

Not applicable.

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
