# Peer review of "Pharmacokinetics and Clinical Implications of Two Non-Tablet Oral Formulations of L-Thyroxine in Patients with Hypothyroidism"

_jcm, 2022, doi:10.3390/jcm11123479_

Round 1

Reviewer 1 Report

The paper presents literature review as a summary of pharmacokinetic, drug and food interactions and clinical data focusing on new oral formulations of L-T4, liquid and soft-gel capsule in healthy volunteers and  patients with primary hypothyroidism. Authors have presented results of 36 papers, (four of them self-citations), with the problems on tablet levothyroxine therapy, with improving results on non-tablet substitution, suggesting  as a conclusion, a non-tablet levothyroxine oral therapy being the first-line choice in treating hypothyroidism especially if suboptimal performance of tablets could be suspected.

The manuscript is clear, and relevant for a great number of patients having problems with not achieving expected concentration and physiological effect of thyroid hormones with tablet L-thyroxin therapy in spite of following prescribed doses, for several causes such as interactions with other drugs, food or not respected fasting conditions necessary for appropriate adsorption. The manuscript is composed as a review from the literature divided in six parts: 1.Introduction, 2.General & pharmacological considerations, 3.Bioequivalence studies with soft gel capsule and oral liquid formulation, 4.Switching from the usual tablet to solution or soft gel capsule formulations in clinical practice, 5.The oral solution and soft gel capsule formulations display better pharmacokinetic profiles than the usual tablets of levothyroxine in case of drug-drug or drug-food interactions, and 6.Conclusion which summarizes the statement of several clinical studies  that the non-tablet L-T4 soft gel capsules and oral solution have proven bioequivalence with the usual L-T4 tablets princeps and generic formulations using 90% CI% of the systemic exposure geometric mean ratios, even suggested higher performance of non-tablet formulations than  tablet in those patients with suboptimal adherence to the correct drug ingestion, the impact of gastrointestinal conditions and variation of gastric pH has been demonstrated to  be lower with either the soft gel capsule or oral solution than with the tablet, the  extent of drug-drug and drug-food interactions remain low and of uncertain clinical relevance. Using soft  gel and liquid L-T4  oral formulations have improving a treatment that needs  to be individually  tailored, and we should  consider  using non-tablet L-T4  formulations as first-line choice, especially in those patients with moderate-to-high potential of suboptimal performance of usual tablet as well as in those with concomitant medication increasing the risk for drug interactions and side effects, or gastro-intestinal condition leading to decreased L-T4 intestinal absorption.

The cited 36 references are publications within the last five years in order as they appear in text, with four of them being self-citations.

The manuscript is composed as a review from the literature already published underlying the use of liquid and soft-gel capsule, a non-tablet L-T4 therapy being more effective in patients with difficulties with absorption.

There are two tables in the manuscript both without figers, showing
1.Pharmacokinetics of levothyroxine in Kidney impairment, Cirrhosis, Elderly, Children, Pregnancy, Gastrointestinal disorders,  Food interaction (adapted from reference #5),
2.
Impact of drug and food on levothyroxine absorption (references 5, 24).

Conclusions are summarized from several cited and reviewed clinical studies.

Author Response

We really thank this Reviewer who appreciates our paper.

Reviewer 2 Report

Finding the individually tailored levothyroxine (LT4) dose is still challenging. Alternative non-tablet oral formulations of LT4 are improving the treatment of hypothyroidism and a literature review in this field represents a relevant topic.

Following suggestions are reported:

-      - Authors reported that clinical studies have suggested higher performance of non-tablet formulations than tablet in those patients with suboptimal adherence and also focused on the impact of gastrointestinal conditions, drug-drug or drug-food interactions on the absorption of different LT4 formulations. Neverthless, authors reported, at line 169, that in a smaller clinical study involving only 18 patients without intestinal malabsorption, switch from the usual tablet formulation to the new soft-gel capsule at unchanged dose improved TSH serum concentration and balance after 3 months treatment.

In this field it is important to cite two studies by Di Donna V et. al (Endocrine, 2018;59:458-460 and Thyroid, 2014;24:1759-64), in which authors identified the major predictive factors of LT4 requirement and showed that LT4 requirement was not significantly different between soft gel capsules and tablets in 103 patients without malabsorption underwent total thyroidectomy, but TSH was significantly lower with soft gel capsules, without a significant concomitant increase in FT4 levels. Considering that the model of post-surgical hypothyroidism should not have significant bias, this finding seems relevant and must be considered in clinical practice, particularly for patients with TSH values close to the limits of the desired therapeutic range or in patients in whom a narrower therapeutic TSH goal is required.

-    -Line 120: change “the classical sodium levothyroxine” in “the tablet sodium levothyroxine formulation”

-    - Line 127-128: clarify the sentence “showing proportionality, at a dosage of 600 mg in healthy volunteers to allow sufficient discrimination between endogenous and exogenous L-T4”

-     - Lines 155-158: the main results of the CONTROL Surveillance Project should be better reported; for example, more than 80 % of surveyed patients reported having had at least one change in their prescribed hypothyroid medication since beginning therapy. Many patients (16.0 %) reported 5–10 changes, and 6.1 % reported having >10 changes.

Author Response

Finding the individually tailored levothyroxine (LT4) dose is still challenging. Alternative non-tablet oral formulations of LT4 are improving the treatment of hypothyroidism and a literature review in this field represents a relevant topic.

Following suggestions are reported:

- Authors reported that clinical studies have suggested higher performance of non-tablet formulations than tablet in those patients with suboptimal adherence and also focused on the impact of gastrointestinal conditions, drug-drug or drug-food interactions on the absorption of different LT4 formulations. Nevertheless, authors reported, at line 169, that in a smaller clinical study involving only 18 patients without intestinal malabsorption, switch from the usual tablet formulation to the new soft-gel capsule at unchanged dose improved TSH serum concentration and balance after 3 months treatment.

In this field it is important to cite two studies by Di Donna V et. al (Endocrine, 2018;59:458-460 and Thyroid, 2014;24:1759-64), in which authors identified the major predictive factors of LT4 requirement and showed that LT4 requirement was not significantly different between soft gel capsules and tablets in 103 patients without malabsorption underwent total thyroidectomy, but TSH was significantly lower with soft gel capsules, without a significant concomitant increase in FT4 levels. Considering that the model of post-surgical hypothyroidism should not have significant bias, this finding seems relevant and must be considered in clinical practice, particularly for patients with TSH values close to the limits of the desired therapeutic range or in patients in whom a narrower therapeutic TSH goal is required.

AUTHORS: we perfectly agree with the Reviewer about the need for including and commenting these two papers. Accordingly, we cited and discussed that in the revised manuscript

- Line 120: change “the classical sodium levothyroxine” in “the tablet sodium levothyroxine formulation”

AUTHORS: we have changed this part according to this comment.

- Line 127-128: clarify the sentence “showing proportionality, at a dosage of 600 mg in healthy volunteers to allow sufficient discrimination between endogenous and exogenous L-T4”

AUTHORS: the sentence was indeed unclear and according to reference 14, the sentence was changed accordingly (Lines 126-128)

- Lines 155-158: the main results of the CONTROL Surveillance Project should be better reported; for example, more than 80 % of surveyed patients reported having had at least one change in their prescribed hypothyroid medication since beginning therapy. Many patients (16.0 %) reported 5–10 changes, and 6.1 % reported having >10 changes.

AUTHORS: we agree with the Reviewer and thank him for this suggestion. Accordingly, the results of CONTROL Switch Study (ref 24) were extended and the results of CONTROL Surveillance Project (17) were also more appropriately reported.
